# Influence of Early Life Factors on the Breast Milk and Fecal Microbiota of Mother–Newborn Dyads

**DOI:** 10.3390/microorganisms12112142

**Published:** 2024-10-25

**Authors:** Emmanuel Cervantes-Monroy, Imelda C. Zarzoza-Mendoza, Samuel Canizales-Quinteros, Sofia Morán-Ramos, Judith Villa-Morales, Blanca E. López-Contreras, Fairt V. Carmona-Sierra, Maricela Rodríguez-Cruz

**Affiliations:** 1Laboratorio de Nutrición Molecular, Unidad de Investigación Médica en Nutrición, Hospital de Pediatría, Centro Médico Nacional Siglo XXI, Instituto Mexicano del Seguro Social (IMSS), Ciudad de México 06720, Mexico; emmanuel.cervam@gmail.com (E.C.-M.); ceci.zarzoza@gmail.com (I.C.Z.-M.); jusyvm@hotmail.com (J.V.-M.); 2Posgrado en Ciencias Biológicas, Universidad Nacional Autónoma de México (UNAM), Unidad de Posgrado, Edificio D, 1° Piso. Circuito de Posgrados, Ciudad Universitaria, Ciudad de México 04510, Mexico; 3Unidad de Genómica de Poblaciones Aplicada a la Salud, Departamento de Biología, Facultad de Química, Universidad Nacional Autónoma de México, Ciudad de México 04510, Mexico; cani@unam.mx (S.C.-Q.); smoran@inmegen.gob.mx (S.M.-R.); blopez@inmegen.gob.mx (B.E.L.-C.); 4Instituto Nacional de Medicina Genómica (INMEGEN), Ciudad de México 14610, Mexico; 5Departamento de Alimentos y Biotecnología, Facultad de Química, Universidad Nacional Autónoma de Mexico, Ciudad de México 04510, Mexico; 6Unidad de Medicina Familiar Number 4, Instituto Mexicano del Seguro Social (IMSS), Ciudad de México 06720, Mexico; fairt.carmona@imss.gob.mx

**Keywords:** mom–newborn dyad, gut microbiota, breast milk, newborn sex, age, nutrition status, delivery mode, early-life microbiota

## Abstract

Maternal gut and breast milk (BM) are key in vertically transmission bacteria to infants, shaping their gut microbiota in early life. Although the establishment of early gut microbiota is known, the role of the combined influence of maternal factors and newborn characteristics is not explored. In this study, we aimed to assess the influence of maternal BMI and total body fat, age, delivery mode, and newborn sex on the diversity and composition of the BM and gut microbiota (GM) in mother–newborn dyads. In this cross-sectional study, of the 986 pregnant women candidates, 53 participated, and, finally, 40 mother–newborn dyads exclusively breastfeeding at 20–28 days postpartum were included. Metataxonomic profiling of DNA extracted from BM and fecal samples was conducted using 16S rRNA sequencing. Globally, the findings offer valuable insights that excessive adiposity, age, and C-section delivery influence a lower abundance of specific taxa in the BM, maternal gut, and gut of newborns. Also, the simultaneous analysis of maternal factors and newborn characteristics shows that maternal age and newborn sex explain an important variation in the microbiota composition. These results add to the understanding of the intricate interplay between maternal factors and the microbial communities that influence early-life gut and BM microbiota.

## 1. Introduction

The neonatal period is a crucial stage for the establishment of the gut microbiota (GM), which plays a key role in short- and long-term health outcomes [1]. The first significant exposure to microbiota occurs during birth when a neonate ingests native microorganisms from the maternal vagina and gut [2,3]. Subsequent exposure to breast milk (BM) provides additional bacteria diversity, possibly from several sources, including the maternal skin, newborn oral cavity, or maternal gut via the entero-mammary pathway [4,5]. While previous studies have investigated the influence of factors like maternal nutritional status and delivery mode on GM and BM microbiota diversity [6], evidence remains limited for others like maternal age and newborn sex.

Obesity has been associated with GM imbalance, which could be passed to the BM microbiota composition [7,8,9]. As a result, newborns born to mothers with obesity may inherit a microbiota that is different, with reduced diversity, decreased *Bifidobacterium* abundance, and increased *Staphylococcus* abundance, predisposing them to childhood obesity [10].

New evidence indicates that the age of a mother can influence the composition of the gut microbiota during pregnancy. Women over 35 years old tend to have higher levels of *Prevotella bivia* and opportunistic microorganism, compared to younger women [11]. Furthermore, BM from women over 30 years old tends to display greater microbiota diversity than BM from younger women [12,13]. The method of delivery mode also impacts maternal gut [14] and BM microbiota [15,16]. Additionally, the delivery mode has a well-known impact on the gut microbiota of newborns, with infants born via C-section showing a microbiota profile like that of their mother’s skin microbiota [17].

Recent evidence suggests that sexual dimorphism plays a role in BM microbiota composition and neonatal GM. It has been observed that BM intended for male neonates has lower diversity and a higher abundance of *Streptococcus* bacteria [17,18]. Similarly, male neonates tend to have less diverse GM and higher abundance of bacteria from the Streptococcaceae family [19].

Despite the existing findings, our knowledge about these factors primarily comes from individual studies, with limited information on the effects of maternal age and newborn sex. Considering the complex interaction of these factors, it is important to evaluate them together. Therefore, the aim of the present study is to assess how maternal BMI, total body fat, age, delivery mode, and newborn sex collectively influence the diversity and composition of GM and BM microbiota in mother–newborn dyads. By understanding these relationships, this study aims to provide valuable insights into the factors that determine the composition of maternal and newborn gut microbiota and breast milk microbiota.

## 2. Materials and Methods

### 2.1. Study Design and Population

This cross-sectional study recruited pregnant women from the Instituto Mexicano del Seguro Social (IMSS). Inclusion criteria comprised women aged 18 to 35 years, first-time mothers without chronic diseases, and those experiencing uncomplicated pregnancy. Newborns included were at term delivery (≥37 weeks gestation), weighed ≥2500 g at birth, and were exclusively breastfed. Participants with mastitis symptoms, newborns with clinical conditions, and those using mixed feeding, antibiotic treatment within 15 days before sample collection, or probiotic supplementation during the sampled period were excluded. The study was conducted in accordance with the Declaration of Helsinki, and the National Commission for Scientific Research of IMSS (R-2017-785-055) approved this study. Written informed consent was obtained from each participant prior to sample collection. The flowchart detailing participants’ eligibility assessment and sample analysis is detailed in Figure 1.

Forty participants were stratified based on primarily (1) maternal body index (BMI) at 20–28 days postpartum (20–28DPP) categorized as normal weight vs. overweight/obesity per World Health Organization guidelines [20] and total body fat mass grouped as adequate body fat (<30%) vs. excessive body fat (≥30%). Other clinical variables were also used, like (2) maternal age (young (≤30) vs. mature (>30 years)), (3) delivery mode (vaginal vs. C-section), and (4) newborn sex (female vs. male). The maternal age categories were chosen because currently, there are increasing numbers of women who delay childbearing, and around the world, the mean age of women at the birth of their first child has crossed the 30-years threshold [21,22].

### 2.2. Procedures

During the home visit, the mother’s body weight and body composition were assessed by bioelectrical impedance using a scale (BC-585F FitScan, TANITA, Arlington Heights, IL, USA) as described elsewhere [23]. Pregestational BMI was calculated using the self-reported body weight (kg) divided by the height (m) squared. The BMI at 20–28DPP was obtained using the body weight measured during the visit. The weight gain during pregnancy was calculated based on the self-reported last gestational weight. Newborn body weight and length measurements were performed supine, without clothes and diapers, using an electronic baby scale and an infantometer (Seca 334, Mobile digital baby scale, Hamburg, Germany) on a flat, solid, and even surface, as recommended by WHO guidelines. The body weight gain from birth (WGFB) and length gain from birth (LGFB) were calculated as the difference between the day of birth and 20–28 postnatal days (20–28PND) data. Finally, the head circumference was assessed using an ergonomic circumference measuring tape (Seca 201) placed around the most prominent area of the forehead, above the eyebrows and ears, and behind the occipital protuberance. All measurements were performed in triplicate.

A home visit was scheduled between 20 and 28DPP, during which participants completed a questionnaire providing demographic data. BM was expressed under aseptic conditions using an electric breast pump (Medela Lactina 0162011, Medela, McHenry, IL, USA), with nipple and areola cleaning with a 0.5% chlorhexidine solution (Famicare, Laboratorio Boniquet de México, Naucalpan Estado de México, México) before expression. The BM samples from both breasts were collected in sterile glass bottles after mixing; a 15 mL aliquot was transferred into a sterile conical tube (Axygen Scientific, Union City, CA, USA) and stored at 4 °C for transportation. Fecal samples were self-collected by participants using an OMNIgene Gut kit (DNAgenotek, Ottawa, ON, Canada) and stored at ambient temperature. At the laboratory, both BM and fecal samples were frozen at −80 °C until microbiota composition analysis.

During the home visit, trained clinical personnel conducted anthropometric measurements of the mother and newborn, as detailed in the Appendix A.

### 2.3. 16S rRNA Gene Sequencing and Data Processing

DNA extraction from BM and fecal samples was followed by amplification of the V4 region of the bacterial 16S rRNA gene and sequencing on the Illumina MySeq 2 × 250-bp platform.

#### 2.3.1. DNA Extraction from Fecal and BM Samples

For DNA isolation from fecal samples (0.2 g) and BM (3 mL), the commercial PowerFecal kit (QIAamp^®^ Power Fecal^®^ DNA, QIAGEN, Germantown, MD, USA) was used following the manufacturer’s instructions, including a mechanical disruption by FastPrep (MP Biomedicals; Solon, OH, USA). The DNA was eluted in a final volume of 100 µL and stored at −70 °C for further analysis. For BM DNA extraction, the samples (3 mL) were thawed to 5 °C and centrifuged at 20,000× *g* for 20 min at 4 °C (Allegra X-22, Beckman Coulter Co., Brea, CA, USA). After that, the fat layer was removed, and the supernatant was discarded. Finally, according to the manufacturer’s protocol, the genomic DNA was isolated from the pellet using the aforesaid commercial kit. The purified DNA was eluted in a final volume of 50 µL and stored at −70 °C for further analysis. The extracted genomic DNA concentration from fecal and BM samples was measured by spectrophotometry (Nanodrop 2000c, Thermo Scientific, Wilmington, DE, USA). The A260/280 ratio was used as a parameter of DNA purity.

#### 2.3.2. PCR Amplification and 16S rRNA Sequencing

The metataxonomic profiling of the fecal and BM microbiota was carried out by the 16S ribosomal RNA (16S rRNA) gene sequencing technique described by López-Contreras et al. [24]. The hypervariable region 4 (V4) was amplified using the oligonucleotides recommended by the Earth Microbiome Project (515F-806R). Briefly, the first PCR amplification was performed with 100 ng of DNA using Platinum TM Taq DNA Polymerase (Invitrogen TM, Thermo Scientific, Carlsbad, CA, USA) under the following PCR conditions: 94 °C for 3 min, 30 cycles of 94 °C for 30 s, 50 °C for 30 s, and 68 °C for 1.5 min, and, finally, 68 °C for 5 min. The DNA integrity was verified by agarose gel electrophoresis (2%). The amplicons were purified through the Agencourt AMPure XP beads (Beckman Coulter, Fullerton, CA, USA) using the Agilent Bravo Automated Liquid Handling Platform (Agilent Technologies, Santa Clara, CA, USA). A second PCR was performed to attach the Illumina sequencing adapters to the amplicons with the following conditions: 94 °C for 2 min, five cycles of 94 °C for 30 s, 60 °C for 30 s, and 68 °C for 1.5 min, and, finally, 68 °C for 5 min. The amplicons were purified, and the libraries’ quantity was assessed using a Qubit^®^ dsDNA HS assay kit on the Qubit 2.0 fluorometer (Life Technologies, ThermoFisher Scientific, Carlsbad, CA, USA). The final library size and concentration were determined by an Agilent D1000 ScreenTape for 4200 TapeStation System (Agilent Technologies, Waldbronn, Germany) and a Qubit 2.0 fluorometer, respectively. Finally, the amplicons were pooled in equimolar ratios and sequenced using the Illumina Miseq 2 × 250-bp platform at the core unit of Instituto Nacional de Medicina Genómica. Negative controls included sterile DNA-free water for fecal and BM DNA extraction and sequencing library preparation. A mock community was used as a positive control (Zymo Research Corp, Irvine, CA, USA).

#### 2.3.3. Bioinformatic Analyses

The raw sequences underwent quality control analysis using the FASTQC (version 0.11.9 [25] tool. Then, the reads were processed and analyzed through the QIIME2 pipeline (version 2020.6 [26]). The sequences were demultiplexed, and DADA2 [27] was used for trimming and filtering to remove sequencing errors in chimeric and or singleton reads, obtaining the amplicon sequence variants (ASVs). The ASVs were referenced against the SILVA 138 database (version 123 [28]) for the 99% sequence similarity taxonomic assignment. Afterward, the metadata, ASV, and taxonomic tables were imported into R software (version 4.1.2 [29]) through the Qiime2R package (version 0.99.6 [30]). The diversity, structure, and taxonomic composition analysis were conducted in Rstudio (version 1.4.1717 [31]) using the phyloseq (version 1.38.0 [32]) and microbiome (version 1.16.0 [33]) packages. Other packages, such as Tidyverse (version 1.3.1 [34]) and Vegan (version 2.6-2 [35]), were also used. A phylogenetic tree was created using the Ape package (version 5.6-2 [36]). The *Decontam* package (version 1.14.0 [37]) used the prevalence method to detect and remove environment or reagent contaminant sequences. Moreover, quality filtering was applied, excluding all samples with less than 5000 reads, ASVs without reads, and those belonging to domains other than Bacteria and Archaea. Core microbiota, which refers to microbial taxa occurring above a particular occupancy frequency threshold, is typically defined using a prevalence threshold ranging from 30% to 95% [38]. This study used a prevalence threshold of 70% to identify the core microbiota from maternal and newborn fecal and BM samples. The alpha diversity (within-sample diversity) was estimated with the Chao (richness) and Shannon (diversity) metrics, rarefying the sequences to 5000 reads. The unweighted and weighted UniFrac distances were calculated, and a principal coordinate analysis (PCoA) was performed to assess the dissimilarity in microbiota community structure (β diversity) between samples. A differential abundance test was conducted, employing the linear discriminant analysis (LDA) effect size (LefSe [39]) tool to identify genus-level differences between groups based on maternal nutrition status and age, delivery mode, and newborn sex.

### 2.4. Statistical Analysis

Statistical analysis was performed using R software (version 4.1.2 [29] in the RStudio environment (version 1.4.1717 [31]). The Gaussian distribution of anthropometric parameters and alpha diversity were computed using the Shapiro–Wilk test. Alpha diversity between groups was compared using parametric T-test or nonparametric Mann–Whitney U-test. Significant differences in beta diversity between groups were determined by permutational multivariate analysis of variance (PERMANOVA) with 1000 permutations. Variation of community structure explained by the maternal and newborn characteristics was depicted using the Vegan package envfit function. According to Lefse analysis, genera with an LDA score above a threshold of 2.0 were considered differentially abundant. Statistical significance was set at a *p* < 0.05.

Eleven participants in each group of normal weight or overweight/obesity were estimated to provide 80% study power to identify a difference of 11.2% of abundance in Firmicutes with an assumption of a standard deviation (SD) of 9.2% with an α-value = 0.05 according to the paper of Verdam et al. [40]. However, we decided to augment the sample size and included more lactating women to reach almost twice the calculated sample size and to explore the data analysis.

## 3. Results

### 3.1. Flowchart of the Participants

A total of 986 pregnant women participated in the Program of Educational Strategies for Health during Pregnancy and Lactation between November 2018 and March 2020, of which 40.87% decided to participate (Figure 1). During the telephone follow-up and home visit at 20–28DPP, 86.85% of the candidates were excluded due to failing to meet the inclusion criteria or refusing to continue their participation. Finally, fifty-three mother–newborn dyads were enrolled in the study. However, 13 dyads were excluded due to technical issues with the maternal fecal, BM, or newborn fecal samples. These samples comprised 20 mother fecal–BM–newborn fecal triads, 15 mother fecal–newborn fecal pairs, two mother fecal–BM pairs, one BM–newborn fecal pair, one mother fecal sample, and one BM sample for data analysis.

### 3.2. Demographic Characteristics

Table 1 summarizes the demographic characteristics of the participants. The median age at enrollment was 29.6 years, with a mean pregnancy body weight gain of 10.51 kg. According to BMI, over half of the participants were classified as overweight or obese before pregnancy and at 20–28DPP. However, considering total body fat, a larger proportion (72%) of women exhibited excessive body fat at 20–28DPP. Vaginal delivery occurred in approximately 53% of the mothers, with female newborns accounting for 45% of the total sample.

### 3.3. Microbial Composition

The maternal gut microbiota comprised 15 distinct phyla, with Firmicutes and Bacteroidota accounting for 91.7% of the reads; minor phyla included Actinobacteriota, Proteobacteria, and Verrucomicrobiota, among others (Figure 2A and Appendix A). Out of the 304 genera detected, those belonging to the Firmicutes phylum were most prevalent (Figure 2B and Appendix A). The maternal GM core included genera such as *Blautia*, *Bacteroides*, *Faecalibacterium*, *Dorea*, *Anaerostipes*, *Fusicatenibacter*, *Bifidobacterium*, *Parabacteroides*, *Coprococcus,* and *Escherichia*-*Shigella*.

The BM microbiota comprised 23 phyla, with Firmicutes dominant, followed by Proteobacteria, Actinobacteriota, and Bacteroidota (Figure 2C and Appendix A). Among the 402 genera that comprised the BM microbiota (Figure 2D and Appendix A), the most abundant were *Streptococcus* and *Staphylococcus*, both belonging to the Firmicutes phylum. The BM core included *Streptococcus*, *Staphylococcus*, *Escherichia*-*Shigella*, *Bifidobacterium*, and *Gemella*.

The newborn GM was primarily made up of the phyla Proteobacteria, Actinobacteriota, Firmicutes, and Bacteroidota, comprising 99.95% of the composition (Figure 2E and Appendix A). The most abundant genera were *Bifidobacterium*, *Escherichia*-*Shigella*, *Pseudomonas*, and notably other genera from the Proteobacteria phylum, with abundances under 1%, but collectively comprising 18.26% of the total composition (Figure 2F and Appendix A). The neonatal GM core detected consisted of *Bifidobacterium*, *Escherichia*-*Shigella*, *Streptococcus*, and *Staphylococcus*.

### 3.4. Factors Influencing Mother–Newborn Gut and BM Microbiota Composition

#### 3.4.1. Maternal BMI and Total Body Fat

Maternal BMI did not significantly affect the richness and diversity of the mothers’ and newborns’ GM and BM microbiota. Nevertheless, mothers GM with excessive adiposity had a lower abundance of certain genera, such as *Fusobacterium*, along with a higher abundance of bacteria belonging to the Firmicutes phylum (Figure 3A). Moreover, BM microbiota from mothers with excessive adiposity (Figure 3B) displayed a lower abundance of specific genera belonging to the Proteobacterium phylum, accompanied by a diminished abundance of fatty-acid-producing bacteria like *Faecalibacterium*, *Anaerostipes,* and *Butyrivibrio*. Furthermore, we notice that newborns born to mothers with excessive adiposity (Figure 3C) showed a lower abundance of core member *Staphylococcus*.

#### 3.4.2. Maternal Age

No differences in the richness and diversity of maternal GM and BM microbiota were found. However, a significant clustering in maternal GM was noted using the weighted UniFrac distance (*R*^2^ = 0.09, *p* = 0.01). Mature women exhibited a distinct microbial profile with enrichment of core member *Bacteroides* and other genera belonging to Bacteroidota phylum, while younger women showed enrichment of certain genera from the Firmicutes phylum, including *Clostridium sensu stricto 1* and *Roseburia*, among others (Figure 4A). Subtle changes in BM microbiota were observed, with a higher abundance of minority genera like lactic acid bacteria *Leuconostoc* and the archaeal *Methanobrevibacter* in BM from mature women (Figure 4B). Notably, newborns GM from mature mothers showed lower richness (*p* = 0.01) and a trend towards lower diversity (*p* = 0.06) compared to neonates from younger women. Although no distinction between beta diversity was observed, the GM of newborns from mature women displayed a lack of enrichment in any taxa and instead had a decreased abundance of Bacteroidales order, including the prevalent genus *Bacteroides,* compared to those from younger mothers (Figure 4C).

#### 3.4.3. Delivery Mode

While delivery mode did not affect the richness, diversity, or overall structure of the maternal GM and BM microbiota, women who had undergone cesarean delivery were associated with specific alterations in maternal and BM microbiota composition. Women who underwent C-section exhibited a lower abundance in genera from the *Ruminococcaceae* family, accompanied by an enrichment observed in a minor genus such as *Erysipelatoclostridium* compared to those women with vaginal delivery (Figure 5A). In addition, BM from women with C-section delivery showed a reduced abundance of the minority genera belonging to the Firmicutes (Figure 5B). Regarding newborn’s GM, while delivery mode did not affect richness, it influenced diversity (*p* < 0.01) and the community structure (unweighted UniFrac: *R*^2^ = 0.04, *p* = 0.01 and weighted UniFrac: *R*^2^ = 0.08, *p* < 0.01). While GM of C-section newborns indicated higher abundances of bacteria from order Oscillospirales, those born vaginally exhibited a higher abundance of bacteria from the *Lachnospiraceae* family (Figure 5C).

#### 3.4.4. Newborn Sex

Newborn sex did no impact alpha diversity of maternal gut, BM, or newborn gut microbiota. However, differential analysis revealed distinct microbial profiles. The maternal GM of female newborns displayed higher abundance in different minority genera, such as *Stenotrophomonas* and *Paludicola* (Figure 6A). In BM samples from mothers with male newborns, there was an enrichment in the core member *Bifidobacterium* (Figure 6B), among others. Finally, we identified differences in composition between females and males in their gut microbiota (Figure 6C).

### 3.5. Exploring the Impact of Maternal and Neonatal Factors on GM and BM Microbiota

Multivariate analysis revealed significant associations between maternal and neonatal factors and the composition of gut and BM microbiota. Maternal age emerged as a key determinant, contributing to 20.9% (*p* = 0.022) and 14.4% (*p* = 0.075) of the variation in maternal GM community, as demonstrated by unweighted and weighted UniFrac analyses, respectively (Table 2). Furthermore, maternal age explained 19.4% (*p* = 0.038) of the total variation in neonatal gut microbiota structure as determined by unweighted UniFrac distance; delivery mode also contributed to 16.3% of the variability but did not reach statistical significance (*p* = 0.060). Notably, the newborn sex accounted for 29.3% of the variation in the BM microbiota community (*p* = 0.028).

## 4. Discussion

Our findings align with previous evidence linking the mother’s total body adiposity and delivery mode to changes in the GM composition in both mothers and newborns, as well as BM. Furthermore, our study highlights the impact of maternal age and newborn sex on the GM of mother–newborn dyads and BM, respectively. These findings represent a significant advancement in our understanding of how various factors influence microbiota in the maternal–infant context, providing new perspectives for future research.

Our study aligns with previous research, revealing that maternal GM reflects typical adult composition [41], and sheds light on postpartum women’s GM, a less explored area [42]. Notably, differences in lactating and nonlactating women’s GM imply persistent post-pregnancy effects [43,44]. Our findings suggest a potential GM imbalance persisting between 20 and 28 days postpartum, possibly aiding microbial translocation to mammary glands [45,46]. The wide variety of genera within the Firmicutes phylum, particularly during lactation, may support maternal gut health via immune stimulation and reduced inflammation through short-chain fatty acids (SCFAs) and lactate production [47,48]. While the lactating women’s GM core remains unclear, certain genera identified are common in healthy adults [49] and may influence gamma-aminobutyric acid production [50], linked to postpartum depression risk reduction [51].

In our study, despite variations in abundance among BM samples from participants, identified taxa are in line with the findings of most studies [15]. Our findings support the notion that breastfeeding provides more than just nutrition; it serves as a natural reservoir of bacterial signatures that are beneficial for newborns’ gastrointestinal and immune system development [52,53].

In neonates, the GM displayed bacterial signatures indicative of early life [54]. Given that the Proteobacteria phylum includes a wide variety of Gram negative potential pathogenic bacteria [55], their presence in the neonatal GM may reflect an evolutionary strategy aimed at stimulating the immune system, increasing its tolerance, and preventing the overgrowth of gut pathogens [56]. In addition, we observed that *Bifidobacterium* was notably represented alongside bacteria from the *Enterobacteriaceae* family. This family can create anaerobic conditions [57], allowing the settlement of strictly anaerobic bacteria like *Bifidobacterium*, *Clostridia*, and *Bacteroides* [1]. *Bifidobacterium*, as a core member in newborn GM, plays a vital role in metabolizing milk oligosaccharides and promoting the maturation of the gastrointestinal and immune system [58].

Our findings demonstrated evidence of the interplay between perinatal factors and the microbiota profiles of gut mothers, BM, and gut newborns. For instance, women with excessive body fat showed alterations in the abundance of several bacterial genera, primarily belonging to the Firmicutes phylum. These changes could potentially influence the host’s energy balance [59] and contribute to a persistent dysbiotic state, with implications for the intergenerational transmission of obesity [60]. Furthermore, women with obesity exhibited a lower abundance of *Fusobacterium* in the GM, which contradicts its association with obesity and unhealthy metabolism [61,62]. Considering that the GM during pregnancy resembles that of individuals with obesity or diabetes [63], this supports our hypothesis that maternal GM changes could persist until 28 days after childbirth. Additionally, in light of this, the *Fusobacterium* includes Gram-negative opportunistic anaerobic bacteria [64] that are part of the endogenous microbiota of the oral cavity [65], and that maternal oral microbiota undergo changes during pregnancy, including an increase in the presence of pathogenic bacteria in the oral cavity [66,67]. The lower abundance of *Fusobacterium* in the GM of women with obesity suggests the possibility of an alteration in the oro–intestinal microbiota axis, as both mutually influence each other through microbial transmission [68]. However, further studies are needed to confirm this hypothesis. It is relevant to mention that although the identified taxa represent a minority, it would be interesting to evaluate whether changes in composition due to nutritional status have the potential to modify the functionality of the maternal GM during lactation. Our findings support the limited impact of BMI and total body fat on BM microbiota, as indicated by a recent review [69]. However, our observation of a reduced abundance of certain bacteria within the Proteobacteria phylum, along with decreased levels of SCFAs-producing bacteria exhibited by the BM from women with excessive body fat, are particularly relevant. This reduction may potentially affect the establishment of the newborn GM, particularly in infants born to mothers with obesity, thereby elevating the risk of inflammatory diseases during childhood, such as atopy or childhood overweight, as suggested in the literature [70,71,72,73,74]. When studying the neonatal microbiota, we consistently observed that maternal BMI did not influence diversity and structure, but total body fat did influence the abundance of different taxa, particularly affecting the core member *Staphylococcus*. The decreased abundance of *Staphylococcus* in newborns born to women with elevated adiposity may appear contradictory to previous studies associating *Staphylococcus*, specifically *S. aureus*, with obesity [10,75,76,77]. However, these discrepancies could be attributed to the diverse species within the *Staphylococcus* genus, such as *S. epidermidis*, which is vertically transmitted from mother to newborn through BM [53,78]. Given that *Staphylococcus* is part of both the core microbiota of BM and newborn gut, we suggest further investigation using more specific approaches.

According to maternal age, we observed differences in the gut microbiota between younger and older mothers, which also extended to the gut microbiota of their newborns. Maternal GM exhibited distinct clustering patterns, with an enrichment of genera within the Bacteroidota phylum among mature women. Surprisingly, offspring born to mature women not only displayed a lower richness but also showed a decreased abundance of Bacteroides, contrary to expectations based on maternal profiles. This reduction was evident among other members within the Bacteroidales order, which play crucial roles in immune system development [79,80]. We observe subtle differences in BM, suggesting the possibility that maternal age may influence neonatal gut microbiota independently of BM microbiota. However, we acknowledge studies that have found differences in terms of age [9,12,13]. These distinctions in BM microbiota could be due to several factors, including changes in the mammary gland over a woman’s lifespan [9,81], alongside methodological variations. Therefore, it would be interesting to evaluate whether these changes are due to intrinsic aging processes linked with progressive loss of intestinal and immune homeostasis [82] or factors associated with age such as modifications in diet, social environment, medication use, and decreased physical activity [83,84].

In relation to delivery mode, we noted a decrease in the abundance of certain obligate anaerobes, such as members of the Ruminococcaceae family, in women who underwent C-sections, consistent with previous findings [14]. This alteration could be attributed to abdominal trauma resulting from the C-section procedure, potentially affecting the maternal gut microenvironment, and leading to a reduction in the abundance of anaerobic bacteria. Moreover, disparities were noted between BM samples from women who had vaginal deliveries and those who underwent cesarean sections, with the latter exhibiting lower levels of anaerobic bacteria. Therefore, alterations in the maternal gut environment may influence the composition of less prevalent microbiota members found in BM. While these changes in the gut and BM microbiota may be imperceptible, they undoubtedly impact the neonatal GM. Cesarean section is recognized as a factor capable of disrupting the establishment and development of the GM [17,85]. Our study identified distinct patterns in the abundance of SCFA-producing bacteria, with a lower abundance of *Lachnospiraceae* bacteria in the gut microbiota of neonates delivered by cesarean section, consistent with existing evidence [85]. The altered profile caused by C-section may disrupt microbial community ecology during establishment [86], along with functional repertoires involved in metabolic and immune responses [85,87]. However, the long-term effects of C-sections in our study population need to be assessed, considering the increasing use of elective C-sections. It is crucial to determine whether observed differences are specifically linked to C-sections or related factors such as antibiotic treatment, medications, or exposure to controlled environments [88]. The majority (57.9%) of participants in our study were unaware of the specific type of antibiotic that they were prescribed. Among those who provided information, cefalosporin was the most frequently (26.3%) mentioned antibiotic. Due to the wide range of antibiotic types and limited data, we were unable to conduct a robust statistical analysis of the antibiotics’ impact on the composition of the microbiota. As per the clinical practice guidelines of our country, prophylactic antibiotics are commonly administered during cesarean sections, with cefalosporin being one of the most frequently used due to its broad spectrum and low bacterial resistance.

According to newborn sex, the multivariate analysis allowed us to propose that neonatal sex explains a portion of the variation in the microbiota composition from BM. For instance, BM from women with males showed enrichment in *Bifidobacterium*, a keystone genus involved in GM establishment. This finding supports the hypothesis that BM may be sex-specific and provide additional protection to male newborns in response to “male disadvantage” [89,90], although the underlying mechanism remains unknown. Furthermore, we observed a higher abundance of *Stenotrophomonas*, an efficient estrogen degrader, and *Pladulicolla*, a genus belonging to the *Ruminoccoccacea* family, positively associated with systemic nonovarian estrogen, and related to the ability to metabolize steroids [91,92], in samples from women who had females. Moreover, we identified differential composition between females and males, possibly due to changes related to sex hormones, although these became more evident after puberty [93].

Our study presents the following points to be highlighted and certain limitations. We are among the pioneering studies to conduct a comprehensive assessment of several factors, including maternal BMI and total body fat, age, delivery mode, and newborn sex, and their influence on the diversity and composition of both maternal and newborn gut and BM microbiota simultaneously. We meticulously controlled for variables such as newborns’ gestational age, parity, lactation stage, and feeding mode during our analysis, albeit resulting in the exclusion of a considerable number of mother–newborn dyads. To enhance accuracy, we incorporated total adiposity measurements alongside BMI, recognizing the limitations of BMI as a sole indicator. We also implemented rigorous procedures to ensure the robustness of our methodology. Thorough cleaning and emptying of both breasts were conducted to minimize potential biases in sampling, and stringent quality control measures were applied, including the use of negative controls throughout the sample extraction, library preparation, and sequencing processes to identify and eliminate contaminant reads.

However, limitations include the use of 16S rRNA sequencing, which may introduce technical biases, and the exclusion of other potential microbial niches. Additionally, the sample size is limited, warranting caution in interpretation, and highlighting the need for larger cohorts to validate and expand upon our findings.

## 5. Conclusions

Overall, we provided valuable insights into the determinants of both maternal and newborn gut microbiota as well as BM microbiota composition; mainly, we contributed new evidence highlighting the influence of maternal age and newborn sex. Maternal BMI and total body fat, age, delivery mode, and newborn sex were found to have significant associations with microbial profiles. These findings contribute to understanding the complex interplay between maternal factors and the microbial communities that shape early-life gut and BM microbiota. However, further research is essential to fully elucidate the mechanisms underlying these associations and their long-term implications for interventions or therapeutic strategies targeting maternal and neonatal microbiome. Specifically, future studies should incorporate longitudinal designs and employ more comprehensive characterization of microbial species to unravel the related health outcomes.

## Figures and Tables

**Figure 1 microorganisms-12-02142-f001:**
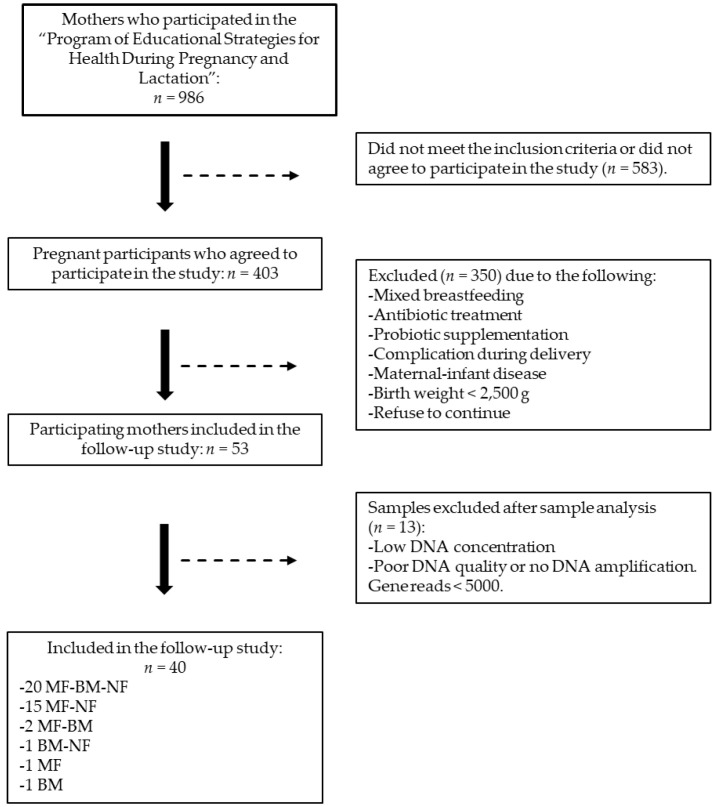
Flowchart of the participants and sample analysis. BM: breast milk, MF: mother feces, NF: newborn feces.

**Figure 2 microorganisms-12-02142-f002:**
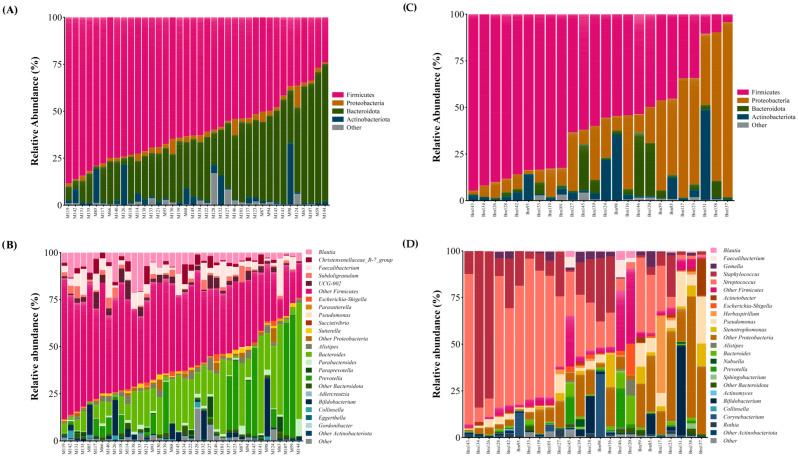
Microbiota composition from the maternal gut at the phylum level (**A**) and at the genus level (**B**), from BM at the phylum level (**C**) and at the genus level (**D**), and from the newborn gut at the phylum level (**E**) and genus level (**F**). Relative abundance bar plot of each sample at the phylum or genus levels. The vertical axis represents the relative abundance, and the horizontal axis is the sample code of the participant. M: mother; Bm: breast milk; Nb: newborn. Genera belonging to the Firmicutes are shown in pink color, Proteobacteria in orange color, Bacteroidota in green color, Actinobacteria in blue, and other distinct genera color are shown in gray color. The newborn GM was primarily made up of the phyla Proteobacteria, Actinomicrobiota, Firmicutes, and Bacteroidota, comprising 99.95% of the composition (Figure 2E and Appendix A). The most abundant genera were *Bifidobacterium*, *Escherichia*-*Shigella*, *Pseudomonas*, and notably other genera from the Proteobacteria phylum, with abundances under 1%, but collectively comprising 18.26% of the total composition (Figure 2F and Appendix A). The neonatal GM core detected consisted of *Bifidobacterium*, *Escherichia*-*Shigella*, *Streptococcus*, and *Staphylococcus*.

**Figure 3 microorganisms-12-02142-f003:**
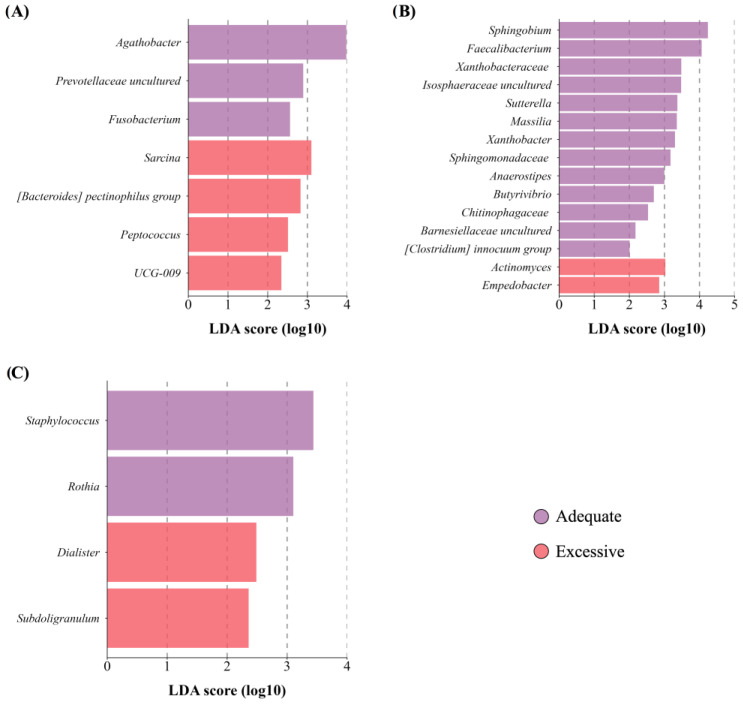
Taxon difference between adequate and excessive maternal total body fat groups. Linear discriminant analysis effect size of maternal fecal (**A**), breast milk (**B**), and newborn fecal (**C**) samples according to maternal total body fat. Pink color bars represent differentially abundant taxa in the adequate adiposity group, while purple color bars represent differentially abundant taxa in the excessive maternal adiposity group. Taxa with significant differences and a minimum linear discriminant analysis (LDA) score of 2.0 are shown.

**Figure 4 microorganisms-12-02142-f004:**
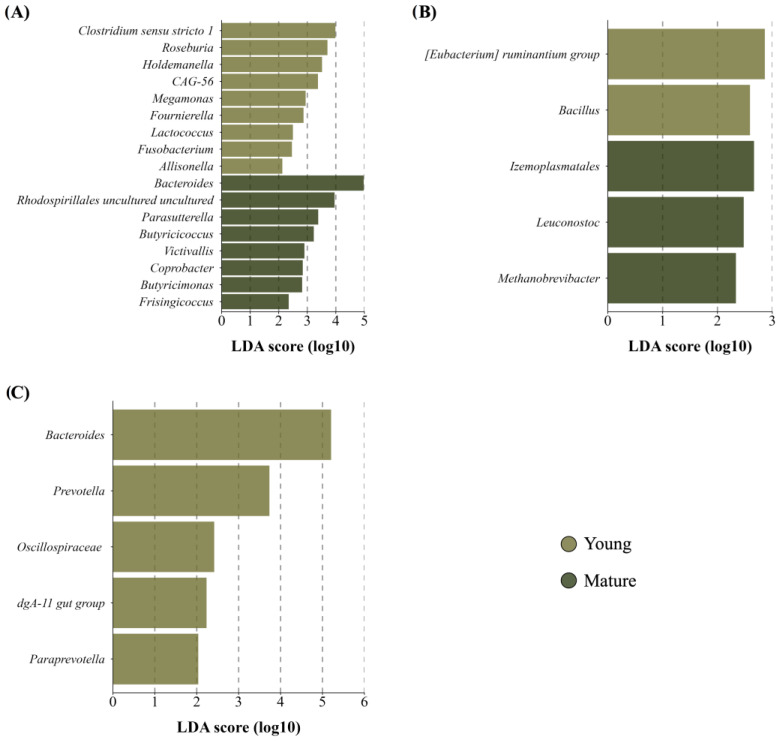
Taxon difference between young and mature age groups. Linear discriminant analysis effect size of maternal fecal (**A**), BM (**B**), and newborn fecal (**C**) samples according to maternal age. Light green bars represent differentially abundant taxa in the young age group, while dark green bars represent differentially abundant taxa in the mature age group. Taxa with significant differences and a minimum linear discriminant analysis (LDA) score of 2.0 are shown.

**Figure 5 microorganisms-12-02142-f005:**
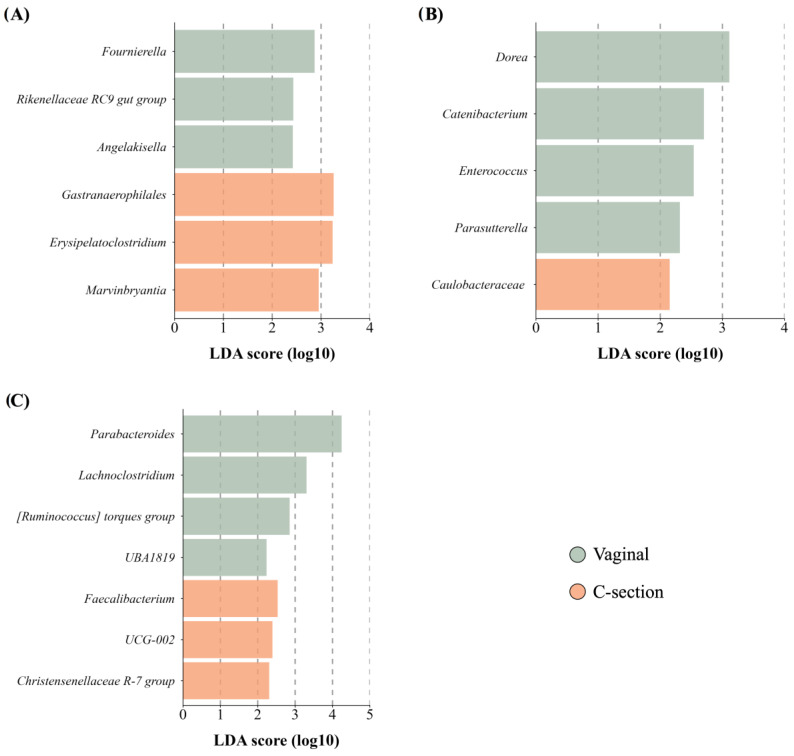
Taxon difference between vaginal and C-section delivery mode groups. Linear discriminant analysis effect size of maternal fecal (**A**), BM (**B**), and newborn fecal (**C**) samples according to delivery mode. Green color bars represent differentially abundant taxa in the vaginal delivery group, while orange bars represent differentially abundant taxa in the C-section delivery group. Taxa with significant differences and a minimum linear discriminant analysis (LDA) score of 2.0 are shown.

**Figure 6 microorganisms-12-02142-f006:**
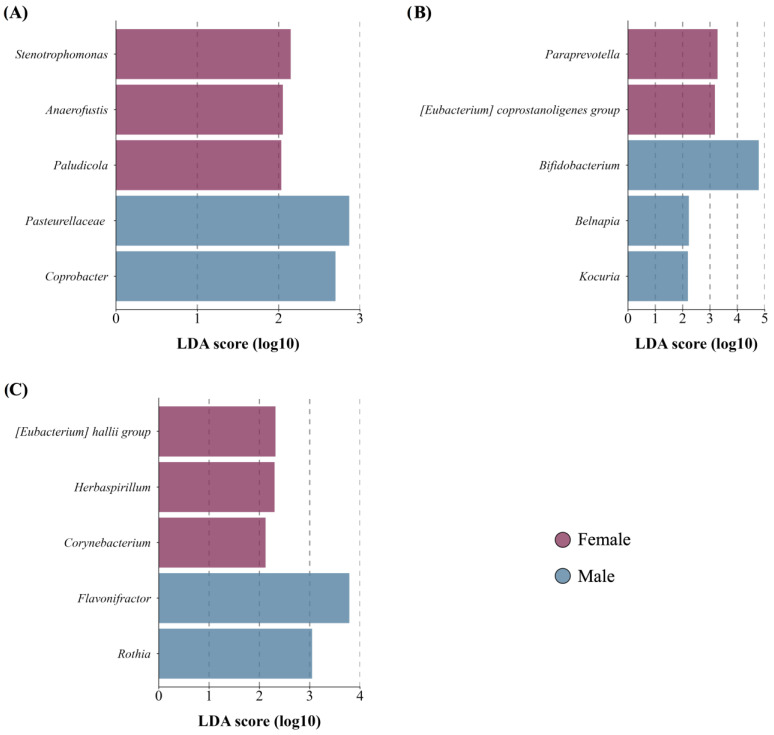
Taxon difference between female and male newborn sex groups. Linear discriminant analysis effect size of maternal fecal (**A**), BM (**B**), and newborn fecal (**C**) samples according to delivery mode. Pink color bars represent differentially abundant taxa in the female newborn sex group, while orange color bars represent differentially abundant taxa in the male newborn sex group. Taxa with significant differences and a minimum linear discriminant analysis (LDA) score of 2.0 are shown.

**Table 1 microorganisms-12-02142-t001:** Demographic characteristics of the mother–newborn dyad participants.

Characteristics	Mean ± SD or Median (Minimum–Maximum)
Maternal	
Age (y)	29.60 (19–35)
Height (m)	1.59 ± 0.06
Body weight (kg)	
Pregestational	63.74 ± 10.51
20–28PPD	64.75 ± 10.64
Last gestational	74.25 ± 10.39
Gain during pregnancy	10.51 ± 4.59
BMI (kg/m^2^)	
Pregestational	25.3 ± 3.99
20–28PPD	25.71 ± 4.07
Total body fat (%)	33.03 ± 6.44
Delivery mode (V/C, %)	53%/47%
Newborn	
Gestational age (weeks)	39.3 ± 1.08
Length (cm)	
At birth	50 (48–53)
At 20–28PND	52.32 ± 1.79
Gain at 20 20–28PND	2.27 ± 1.67
Body weight (kg)	
At birth	3.07 ± 0.33
At 20–28PND	3.81 ± 0.46
Gain at 20–28PND ^1^	0.74 ± 0.47
Head circumference (cm)	36.55 ± 1.01
Newborn sex F/M (%)	45/55

Data were analyzed and expressed as mean ± standard deviation for normally distributed data or median (minimum–maximum) for non-normal distribution data, as appropriate. BMI: body mass index; C: cesarean; F: female; M: male; PND: postnatal day; PPD: postpartum day; SD: standard deviation; V: vaginal; N = 40. ^1^ Gain at 20–28PND = At 20–28PND—at birth.

**Table 2 microorganisms-12-02142-t002:** Envfit analysis on the maternal gut, BM, and newborn GM structure according to UniFrac and weighted UniFrac distances associated with mother–newborn characteristics.

	*R* ^2^	*p*-Value
Mother Gut		
Unweighted UniFrac		
Age, y	0.209	**0.022**
Total body fat, %	0.004	0.944
Delivery mode	0.010	0.830
Sex	0.021	0.707
Weighted UniFrac		
Age, y	0.144	0.075
Total body fat, %	0.002	0.971
Delivery mode	0.033	0.593
Sex	0.041	0.499
Breast milk		
Unweighted UniFrac		
Age, y	0.030	0.719
Total body fat, %	0.016	0.850
Delivery mode	0.163	0.173
Sex	0.139	0.230
Weighted UniFrac		
Age, y	0.086	0.378
Total body fat, %	0.049	0.623
Delivery mode	0.059	0.530
Sex	0.293	**0.028**
Newborn gut		
Unweighted UniFrac		
Age, y	0.194	**0.038**
Total body fat, %	0.004	0.945
Delivery mode	0.163	0.060
Sex	0.019	0.743
Weighted UniFrac		
Age, y	0.048	0.465
Total body fat, %	0.018	0.740
Delivery mode	0.084	0.231
Sex	0.030	0.614

The model was constructed based on PCoA ordination using UniFrac distance for community richness and weighted UniFrac for community abundance. The *R*^2^ represents the proportion of variance explained by ordination. *p*-values are based on 999 random permutations; significant values are in boldface.

## Data Availability

Raw data supporting the findings of the present research will be made available by the authors upon request.

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
