# Peer review of "Influence of Early Life Factors on the Breast Milk and Fecal Microbiota of Mother–Newborn Dyads"

_microorganisms, 2024, doi:10.3390/microorganisms12112142_

Round 1
Reviewer 1 Report
Comments and Suggestions for Authors
A very interesting manuscript investigating factors that Influence the breast milk and fecal microbiota of mother and newborn dyads. The authors started investigated 40 mother-newborn dyads that exclusively breastfed their newborns starting from 986 candidates. This study concluded that adiposity, mother age, delivery by C-section have an influence towards lower abundance of specific taxa in the breast milk, and maternal gut, and newborn gut. Furthermore, mother age and newborn gender can explain an important variation in the microbiota composition.
A few comments that in my opinion will improve this work:
1. Figure S1 is missing from the supplemental material, however as reading the relevant description I propose to have this figure in the manuscript body instead of the supplement.
2. Figure 1, seems that many samples are missing please clarify within the manuscript reasons that there is no information for these samples, furthermore would prefer that at the phylum level, in the horizontal axis the samples are with the same order, thus one can compare, species as found in the different locations.
3. Delivery Mode seems to affect microbiota composition, would be interesting to (if data do exist) investigate the possible role of different antibiotics (again if it is possible to have adequate data) provided to these women in the microbiome. The authors do state in the discussion for the possible role of antibiotics and future assessment however would be nice to perform a mini sub-analysis if data do exist.
Author Response
We thank you for your suggestions because they significantly improved the manuscript.
A very interesting manuscript investigating factors that Influence the breast milk and fecal microbiota of mother and newborn dyads. The authors started investigated 40 mother-newborn dyads that exclusively breastfed their newborns starting from 986 candidates. This study concluded that adiposity, mother age, delivery by C-section have an influence towards lower abundance of specific taxa in the breast milk, and maternal gut, and newborn gut. Furthermore, mother age and newborn gender can explain an important variation in the microbiota composition.
A few comments that in my opinion will improve this work:
Comment 1. Figure S1 is missing from the supplemental material, however as reading the relevant description I propose to have this figure in the manuscript body instead of the supplement.
Response. Thank you for pointing this out. We agree with this comment and we added Figure 1 on page 6, marked in green color, to the manuscript body.
Comment 2. Figure 1, seems that many samples are missing please clarify within the manuscript reasons that there is no information for these samples, furthermore would prefer that at the phylum level, in the horizontal axis the samples are with the same order, thus one can compare, species as found in the different locations.
Response. We agree. We have added the information on page 5, lines 217-227 of the 3.1. Flowchart of the Participants in the Results section. This information is marked in green color.
Comment 3. Delivery Mode seems to affect microbiota composition, would be interesting to (if data do exist) investigate the possible role of different antibiotics (again if it is possible to have adequate data) provided to these women in the microbiome. The authors do state in the discussion for the possible role of antibiotics and future assessment however would be nice to perform a mini sub-analysis if data do exist.
Response. We appreciate the comment. We have accordingly emphasized that the majority (57.9%) of participants in our study needed to be aware of the specific type of antibiotic that they were prescribed. Among those who provided information, cefalosporin was the most frequently (26.3%) mentioned antibiotic. Due to the wide range of antibiotic types and limited data, we were unable to conduct a robust statistical analysis of the antibiotics’ impact on the composition of the microbiota. As per the clinical practice guidelines of our country, prophylactic antibiotics are commonly administered during cesarean sections, with cefalosporin being one of the most frequently used due to its broad spectrum and low bacterial resistance. This information is described on pages 7 and 8 in lines 495 to 502 in the Discussion section.
Reviewer 2 Report
Comments and Suggestions for Authors
This study aimed to assess the influence of maternal BMI and total body fat, age, delivery mode, and newborn sex on the diversity and composition of the BM and gut microbiota (GM) in mother- newborn dyads. This is quite interesting research due to it’s multiple consideration of the factors and get an amazing results. I recommend accepting that after a major revise.
1. The introduction of is not sufficient to introduce the research purpose.
2. What’s the volume and weight were used to extract DNA in section 2.3?
3. Please clarify the PCR primer, condition, system in section 2.3.
4. How you confirm the DNA concentration and purification of the PCR production/
5. Section 2.4, what’s the name of packages author used to analyze the data?
6. Section 2.4, How to deal with the raw data of the sequence?
7. The figures in the main text are not clear enough to read. Please revise them.
Author Response
We thank you for your suggestions because they significantly improved the manuscript.
This study aimed to assess the influence of maternal BMI and total body fat, age, delivery mode, and newborn sex on the diversity and composition of the BM and gut microbiota (GM) in mother- newborn dyads. This is quite interesting research due to it’s multiple consideration of the factors and get an amazing results. I recommend accepting that after a major revise.
Comment 1. The introduction of is not sufficient to introduce the research purpose.
Response. We appreciate your comment and have improved the introduction to better explain the research purpose. The changes are marked in green on pages 1 and 2 pages of the manuscript.
Comment 2. What’s the volume and weight were used to extract DNA in section 2.3?
Response. We apologize for omitting information from the main manuscript. We have added this information on page 3, line 133.
Comment 3. Please clarify the PCR primer, condition, system in section 2.3.
Response. We appreciate the information and have added the “PCR Amplification and 16S rRNA Sequencing” section with the PCR primer, condition, and system on pages 3 and 4, lines 146-168 of a new manuscript version.
Comment 4. How you confirm the DNA concentration and purification of the PCR production/
Response. Thank you for your comment. We have added the information in the “DNA extraction from fecal and BM samples” section on page 3, lines 142-144.
Comment 5. Section 2.4, what’s the name of packages author used to analyze the data?
Response. We apologize for omitting this information from the main manuscript and have added the “Bioinformatic analysis” section with the procedures realized for the data analysis.
Comment 6. Section 2.4, How to deal with the raw data of the sequence?
Response. We appreciate the comment. The raw data are also explained in the “Bioinformatic analysis” section.
Comment 7. The figures in the main text are not clear enough to read. Please revise them.
Response. We agree, and the resolution of the figures has been improved.